# *Puccinia striiformis* f. sp. *tritici* Exhibited a Significant Change in Virulence and Race Frequency in Xinjiang, China

**DOI:** 10.3390/jof10120870

**Published:** 2024-12-14

**Authors:** Hong Yang, Muhammad Awais, Feifei Deng, Li Li, Jinbiao Ma, Guangkuo Li, Kemei Li, Haifeng Gao

**Affiliations:** 1College of Agriculture, Xinjiang Agricultural University, 311 Nongda East Road, Urumqi 830052, China; yh2201832022@163.com; 2Institute of Plant Protection, Xinjiang Academy of Agricultural Sciences/Key Laboratory of Integrated Pest Management on Crop in Northwestern Oasis, Ministry of Agriculture and Rural Affairs, Urumqi 830000, China; xjzbskg@163.com (F.D.); lgk0808@163.com (G.L.); 3State Key Laboratory for Crop Stress Resistance and High-Efficiency Production, College of Plant Protection, Northwest A&F University, Xianyang 712100, China; awaismuhammad@nwafu.edu.cn; 4State Key Laboratory of Desert and Oasis Ecology, Key Laboratory of Ecological Safety and Sustainable Development in Arid Lands, Xinjiang Institute of Ecology and Geography, Chinese Academy of Sciences, Urumqi 830000, China; lili.bobo@outlook.com (L.L.); majinbiao@ms.xjb.ac.cn (J.M.)

**Keywords:** wheat stripe rust, race identification, virulence factor, Xinjiang epidemic region

## Abstract

Xinjiang is an important region due to its unique epidemic characteristics of wheat stripe rust disease caused by *Puccinia striiformis* f. sp. *tritici*. Some previous studies on race identification were conducted in this region, but it is still unclear how temporal changes affect the dynamics, diversity, and virulence characteristics of *Pst* races in Xinjiang. To gain a better understanding, we compared the race data from spring and winter wheat crops of 2022 with that of 2021. Our results showed a significant change in virulence frequency in 2022. *Vr10*, *Vr13*, and *Vr19* exhibited an increasing trend, with a frequency of ≥18%, while the maximum decline was observed in *Vr1*, *Vr3*, and *Vr9*, with a frequency of ≤−25%. It was found that *Yr5* and *Yr15* remained effective against Xinjiang *Pst* races. The race diversity increased from 0.92 (70 races out of 345 isolates) to 0.94 (90 races out of 354 isolates) in 2022, with *G22G* being the dominant race group. Race *CYR34* became prevalent in the region in 2022, while the *LvG* grouped was wiped out in 2022, from both summer and winter crop seasons. *HyG* and *SuG* groups showed an overall declining trend. Overall prevalent races showed over-summering and over-wintering behaviors in Xinjiang. The number of new races occurrence frequency increased by 34% in 2022, indicating a potential change in the population structure of *Pst*. It is crucial to introduce newly resistant gene cultivars in this region and to establish rust-monitoring protocols to prepare for any future epidemics.

## 1. Introduction

Disease resistance genes are widely used in agriculture to reduce disease outbreaks and epidemics [1]. However, pathotype composition within populations is thought to evolve and change over time, due to the selective pressure of planting the same varieties containing the same resistant genes [2]. Pathogen populations adapt due to this constant selective pressure from the host through mechanisms such as outcrossing with other genotypes and mutations.

Wheat is one of the most widely grown food crops globally, and its yield is affected by wheat stripe rust caused by *Puccinia striiformis* f. sp. *tritici* (*Pst*). So far, 86 resistant genes (*Yr1*–*Yr86*) have been identified. However, the effectiveness of these resistant genes in wheat may be overcome through mutation, heterokaryosis, and sexual reproduction [3,4,5]. High genetic diversity of *Pst* was reported in the northwestern region of China [6], and most of the rust-resistant wheat varieties first lost their resistance in this place [7].

In Xinjiang, both spring and winter wheat crops are grown, providing a host for *Pst* throughout the year. Wheat has been grown annually in 0.60–0.75 million hectares since 2000 in Xinjiang, in which there are three different wheat-growing areas, including the spring wheat-planting area, winter wheat-planting area, and spring and winter wheat mixed-planting area [8,9]. Additionally, this region has different barberry species, which are important for *Pst* genetic diversity [10]. However, it is still unclear how the adaptation of new and old *Pst* races occur in relation to the winter and spring crops.

Xinjiang wheat stripe rust epidemic areas are relatively independent in terms of disease outbreaks. Its geographical and ecological characteristics set it apart from other inland epidemic regions in the country [11,12]. The genetic makeup of the Xinjiang population is distinct from that of other epidemic regions in China [6,12]. Due to the different genetic structure, the *Pst* evolution of rust in this region is significantly slower, compared to other inland epidemic regions [8,12,13].

In Xinjiang, wheat rust management programs have mainly relied on the effective use of fungicides and the deployment of R-genes against virulent *Pst* races [14]. However, recent field surveillance conducted in this region has demonstrated the partial ineffectiveness of fungicide applications. Some field farmers used fungicide but still experienced rust. This was also confirmed by Zhou et al. [15], who reported that newly emerged races in the Ili region of Xinjiang have shown resistance against triadimefon. Natural mutation in *Pst* led to the efficacy loss of triadimefon to control the disease. On the other hand, single-gene resistance is rapidly overcome, due to the frequent occurrence of new virulent fungal strains [16]. It is important to regularly check for rust in this region, as it borders other countries, increasing the chances of rust invasion.

It is important to conduct race evaluation surveys at different timeframes in the Xinjiang region to better understand the complexity and diversity of *Pst*. This will help assess the potential loss of effectiveness of *Yr* genes for controlling wheat rust during this time. Therefore, this study aims to identify changes over time in the complexity of *Pst*, the diversity of races, and the efficiency of *Yr* genes. Understanding how *Pst* races in Xinjiang have evolved and the durability of *Yr* genes will provide valuable insights for guiding wheat breeding and disease management recommendations across the country. The objective of this study is to perform race identification from spring and winter wheat crops in the Xinjiang region during the year 2022 and to compare the overall race and virulence data with 2021 crop season samples.

## 2. Materials and Methods

A total of 354 isolates of wheat stripe rust were collected from winter and spring wheat fields in seven counties, including Qapqal, Yining, Gongliu, Xinyuan, Nilka, Zhaosu, and Tekes (Appendix A) during the year 2022. Sampling points were at least 10 km apart. At each site, 10–15 leaves containing fresh urediniospores were collected and placed into a moisture-absorbing paper bag to dry. Each sample’s location, date, latitude, longitude, and altitude were recorded. The samples were then returned to the laboratory, placed into a drying vessel containing moisture-absorbing silica gel, and stored at −4 °C for later use.

### 2.1. Urediniospores Multiplication and Preservation

Before inoculation, wheat leaves collected containing urediniospores were rinsed with running water, the leaves were unfolded and laid flat, face up, on moistened filter paper, and the sample number was recorded and placed in a dew chamber at a temperature of 10 °C for 8–12 h under dark conditions. The highly susceptible Chinese winter wheat variety Mingxian169 was used as the inoculation material. When the first wheat leaf was fully expanded, a single urediniospores mound was inoculated on the wheat leaf with a sterile large needle pin. After inoculation, the small pot was covered with a plastic cover to avoid cross-contamination, and water was sprayed to moisturize it. Then, it was taken out after 24 h under darkness at 10 °C and then transferred to the growth chamber for cultivation. The light was 16 h/d, and the darkness was 8 h/d for alternation. For inoculation, leaves appeared with faded green spots, with gradual spore production, leaving only one strain of rust per leaf, and the rest of the leaves were cut off; the leaf was placed into a dry glass tube and gently tapped to collect spores. A spore suspension was prepared by adding 3M7100Novec solution to the collected fresh urediniospores, and a pipette inoculation of Mingxian169 seedlings was used for the multiplication of *Pst*. The collected urediniospores were stored in a desiccator in a −4 °C refrigerator, or for long-term storage, they were sealed and placed in a −80 °C ultra-low temperature refrigerator.

### 2.2. Race Identification

The 19 Chinese wheat differential lines carrying single or multiple resistance genes were utilized for the race identification of isolates, as described by Zhan et al. [12]. These 19 different wheat varieties were sown in four corners of plastic pots (approximately 10 cm diameter filled with a potting mix from Inner Mongolian Mengfei Biotech Co., Ltd., Hohhot, China), with 5 grains of wheat placed in each corner, and Mingxian169 was used as a control. About 10 days after planting, the wheat was ready for inoculation when the plant was in the two-leaf stage. Before inoculation, 0.1% of Tween solution was sprayed, and fresh urediniospores were mixed with talc powder at a ratio of 1:30 and were shaken on wheat seedlings. After inoculation, the wheat seedlings were placed on an isolation cover, kept in a dark place for 24 h, and moved for further cultivation to an artificial growth chamber with a day/night thermoperiod of 17/13 °C, a photoperiod of 16 h, and a relative humidity of 60%. About 15 days after inoculation, the symptoms appeared, and the infection type (IT) of each isolated strain on different wheat varieties was recorded when the control, Mingxian169, had sufficient sporulation. Infection types were graded on a 0–9 scale. In this scale, IT was from left to right: 0: no visible disease symptoms (immune), 1: minor chlorotic and necrotic flecks, 2: chlorotic and necrotic flecks without sporulation, 3–4: chlorotic and necrotic areas with limited sporulation, 5–6: chlorotic and necrotic areas with moderate sporulation, 7: abundant sporulation with moderate chlorosis, and 8–9: abundant and dense sporulation without notable chlorosis and necrosis [17].

### 2.3. Data Analysis

Data on different types of infections (virulence/avirulence) on Chinese differentials were recorded and compared with known virulence patterns of races. The data were analyzed using the Virulence Analysis Tool [18] and Excel spreadsheets to calculate the frequency and diversity of virulence. Additionally, cluster analysis using R software (3.6) was performed to assess the relationships among the different races of *Puccinia striiformis* f. sp. *tritici* (*Pst*) in various locations and their virulence characteristics.

## 3. Results

### 3.1. Virulence Frequency of Puccinia striiformis Over Time in Xinjiang

We investigated the virulence factors using a Chinese differential set of known *Yr*-resistant genes and explored how temporal patterns impact the virulence frequency of *Pst* races of spring and winter crops of Xinjiang. The *Pst* samples during the 2022 crop season of spring and winter wheat were analyzed for virulence frequency using 2021 samples of spring and winter wheat [19,20]. The virulences were named as *Vr1*, *Vr2*…… *Vr19* against the Chinese differential order, respectively.

In 2022, the frequencies of virulence in *Vr14*, *Vr2*, *Vr4*, *Vr11*, and *Vr6* were significantly high (>70%). On the other hand, the virulences of *Vr5*, *Vr12*, *Vr1 3*, *Vr17*, and *Vr19* were comparatively low (<50%; Figure 1). No virulences were reported for *Vr15* and *Vr18*, which showed the effectiveness of resistant genes *Yr5* and *Yr15*. However, *Yr15* was reported in low frequency in the 2021 cropping season, and we found a significant change in the overall virulence frequency. The maximum increases were observed in *Vr10*, *Vr13*, and *Vr19* (≥18% frequency), and maximum declines were exhibited in *Vr1*, *Vr3*, and *Vr9* (≤−25% frequency). Upon comparing our 2022 and 2021 results, we found that all virulences, except *Vr15* and *Vr18*, were present during both summer and winter in the Xinjiang region (Table 1).

### 3.2. Evaluation of Different Races from Spring and Winter Wheat Crops on the Temporal Scale

We grouped different races into groups based on their virulence reaction on the Chinese differential set. Comparative analysis results revealed significant variations in the frequencies of different race groups. The largest increase was observed in race group *G22G* (11.33% frequency differences), while the *LvG* and *CYR* race groups were wiped out, and the *SuG* race group showed the most significant decline in 2022 (−22.61% frequency). Additionally, the new race frequencies were increased (34.4%) in 2022, compared to (9.24%) 2021 (Figure 2A; Table 2).

The distribution of races also changed significantly in Xinjiang. The *CYR34* race from the *G22G* group became the most common race in 2022, accounting for 16.1% of the total, compared to 10.01% in 2021. On the other hand, the *Suwon11-1* race showed a decrease in frequency, dropping from 17.17% to 7.3% in 2022 (Figure 2B).

### 3.3. Over-Summering and Over-Wintering of Puccinia striiformis Races in Xinjiang

Further, we investigated whether the predominant races in winter wheat crops can overlap their disease cycle with spring wheat to understand how races survive year-round in the Xinjiang epidemic region.

The *Suwon11-1* race was found at a higher frequency in the 2021 winter crop (18.6%). This frequency slightly declined in the spring crop of the same year (15.74%) and further decreased to 6.6% in the 2022 winter wheat crop. In the spring wheat crop of 2022, it maintained a frequency of 8.1% (Figure 3; Table 3). In contrast, the dominant *CYR34* race displayed over-wintering and over-summering behaviors in both consecutive cropping years. The frequency decreased from 14% to 6.02% from winter to spring in 2021. However, in 2022, the frequency showed an increasing trend, reaching 14% in the winter and further increasing to 18% in the spring wheat crop. The races *Guinong22-13*, *CYR31*, *CYR28*, and *Suwon11-2*, which were over-wintered and over-summered in the 2021 crop season, did not maintain their presence in the 2022 spring and winter crops. The overall result showed that the Xinjiang region could have the potential to over-winter and over-summer the *Pst* race pathogen. However, it also has the potential to generate new virulent races that challenge the breakdown of the resistance of local commercial cultivars.

### 3.4. Race Diversity Within Winter and Spring Wheat Crops on the Temporal Scale

The race diversity showed an increasing trend in the temporal pattern. The race diversity increased from 0.92 (70 races out of 345 isolates) to 0.94 (90 races out of 354 isolates) in 2022. In 2021, race diversity was lower at 0.91 (25 races out of 129 samples) during the winter crop, but it increased in the following spring crop to 0.94 (46 races out of 201 samples). This was followed by a slight increase in the 2022 winter crop, reaching 0.94 (52 races out of 182 samples), followed by the next spring crop at 0.93 (40 races out of 172 samples; Figure 3).

In the 2021 cropping year, the overall newly identified *Pst* race frequency was lower (9.24%) compared to 2022 (34.5%). In the spring 2021 crop, 12.8% new races were identified, which was further increased to 23.6% in the next spring crop of 2022. However, the newly identified race frequency was lower in the 2021 winter crop (5.5%), compared with the 2022 winter crop (36.3%).

### 3.5. Cluster Analysis Based on Pst Groups

The races identified from spring and winter wheat crops in 2022 were organized using Ward’s hierarchical clustering dendrogram based on their virulence and avirulence profiles. *Pst* races from both crops (winter and spring) were clustered into four distinct groups (Figure 4). The maximum number of newly identified races were located in G4, followed by G3. Both clusters included races from the *SuG* and *HyG* groups. The G1 group was only composed of the *CYR34*, *Guinong22-14*, and *Guinong22-11* races, while the other *G22G* races were in G2.

## 4. Discussion

Xinjiang is an important epidemic region for stripe rust disease in China, due to its diverse ecological zone and connectivity with neighboring areas. Some studies were conducted in this region, which focused on race identification. However, the roles of the over-summering and over-wintering of pathogens, as well as the evolution of races over time, remain unclear. Therefore, this study aims to focus on these aspects.

Identifying and comparing races among different regions of the world was challenging, due to the variations in wheat genotypes found within the differential sets [21]. To overcome this challenge, China introduced their own differential set, which contained 19 wheat genotypes with single and multiple *Yr*-resistant genes. This set was implemented at the national level to monitor race virulence structure. So far, different race groups, based on their virulence to specific differential lines, have been identified in China, including *CYR*, *HyG*, *SuG*, and *G22G*. These race groups are found in different epidemic regions of China and have a breakdown of resistance in many famous cultivars [14]. However, limited studies have been conducted on the over-summering and over-wintering of these race groups, especially in the Xinjiang region, where limited research has been performed. In our study, we observed a change in the race frequencies of the Xinjiang region during the years 2021–2022. Some race groups that were previously present [19,20] disappeared, such as the *LvG* group, while the frequency of other groups significantly decreased.

Xinjiang’s *Pst* population is different from those in other epidemic regions of China [22]. However, despite the genetic difference [6], they exhibit some phenotypic similarities because this region has some common races that have also been reported in other regions of China. For example, the race *CYR34*, which is predominant in the Xinjiang region, was initially reported in Sichuan in 2008 [23]. Subsequently, it became prevalent in Gansu in 2014 [24] and other provinces as well [25]. However, based on the population genetic studies conducted thus far, it has been observed that the population genetic structures of Gansu, Sichuan, and other regions of China differ from that of Xinjiang. Therefore, it is imperative to conduct molecular genotyping studies of these races in conjunction with their corresponding phenotypic race data to gain insights into the genetic structures and virulence behaviors of these races [26]. Additionally, it is necessary to incorporate a single *Yr*-gene line into race studies, as it is beneficial for comparing Chinese race virulence information with worldwide data. The Chinese differential lines contain multiple *Yr* genes. These additional resistance genes may potentially restrict the interpretation of results in terms of virulence and avirulence genes [26]. Furthermore, various factors contribute to some of the errors in phenotypic data, including the repeated multiplication of seed stocks of standard differential wheat varieties, which may lead to some degree of intermixing of seeds over time, particularly when plant phenotypes are indistinguishable. Additionally, pathogen isolate contamination may occur, either through sampling plant leaves containing multiple races or due to insufficient laboratory facilities. Disease scores on differentials may vary due to local experimental conditions, such as temperature, humidity, light, and plant nutrition, as well as personal judgments of the virulence phenotype by different researchers [26]. To minimize these errors, a combination of the traditional phenotype race identification method with molecular methods employing a pathogenomics strategy, such as transcriptome sequencing of PST-infected wheat leaves, can be beneficial for understanding the population structure of an emerging pathogen [27].

Xinjiang’s *Pst* population can undergo genetic recombination through sexual reproduction [28], especially after the discovery of an alternative host, barberry [29]. The races that evolved through artificial methods of selfing have the potential to evolve into a more virulent form than the parental race. Similar to the study by Wang et al. [9], we observed that F2 progeny was virulent to *Y5*, while the parental isolate was avirulent on this gene. Our study found an increase in the number of new races in the nature population of Xinjiang, although the population of some old *Pst* race groups decreased. The new races of *Pst* in Xinjiang displayed a complex pattern of virulence. However, the new *Pst* races did not seem to adapt well to local climates, as most of them disappeared in the next cropping season. The fitness of new races may be affected by their genetic characteristics, the resistance genetics of the host, and the environmental factors of survival. In nature in the Xinjiang epidemic area, the genetic characteristics of the pathogen and host are relatively stable, but the environmental conditions are usually quite variable, which may cause new races to be wiped out during their early stage. On the other hand, the old prevalent races, such as *CYR34* and *Suwon11-1*, still maintained their presence, although the frequency of some old races showed a decreasing trend, and some were even wiped out. This decrease could be attributed to the deployment of newly commercial cultivars that may be resistant to old race groups, such as *LvG.*

Cultivating wheat varieties that are resistant to stripe rust is one of the most effective methods for managing the disease and is also environmentally friendly. However, factors such as mutation and somatic hybridization can lead to changes in the virulence of the pathogen population [29]. Selection pressure can favor these mutants or variants, leading to significant changes in the population’s race structure. New races of *Pst* can quickly become more common, causing previously resistant varieties to become susceptible after a few years. Also, climatic changes played a role in the migration of races to warmer areas, as previously, the disease condition was unfavorable in these regions due to low temperature. Effectively managing stripe rust through cultivar resistance and preemptive fungicide application depends on understanding the pathogen’s virulence and genetic structure. A recent study of the *Pst* population showed the migration of *Pst* from central Asia to Xinjiang [30], highlighting the importance of monitoring the *Pst* races in this region.

## 5. Conclusions

This study observed a significant shift in both virulence and race diversity. In 2022, virulence against the differential lines Jubilejina, Kangyin 655, and Guinong 22 increased, compared to the 2021 crop season. Conversely, virulence against Trigo-Eureka, Lutescens 128, and Danish 19 decreased in 2022. Additionally, virulence against Zhong 4 and Triticum spelta Album was absent. This study demonstrated the effectiveness of the *Yr5* and *Yr15* genes. The race groups *CYR* and *LvG* did not maintain their adaptability and disappeared in 2022. Meanwhile, the *G22G* race group exhibited an overall increase in frequency. Race *CYR34* from the *G22G* group has become prevalent in Xinjiang. The number of new races increased in 2022, though only a few new races from 2021 were still cloned in the following year, with no significant increase in their numbers. Most new races are unlikely to maintain their adaptability in the upcoming cropping year, possibly due to extreme climate conditions in Xinjiang. Furthermore, Xinjiang’s proximity to Central Asian countries warrants regular monitoring for any invading races, which would help prompt responses to potential future epidemics.

## Figures and Tables

**Figure 1 jof-10-00870-f001:**
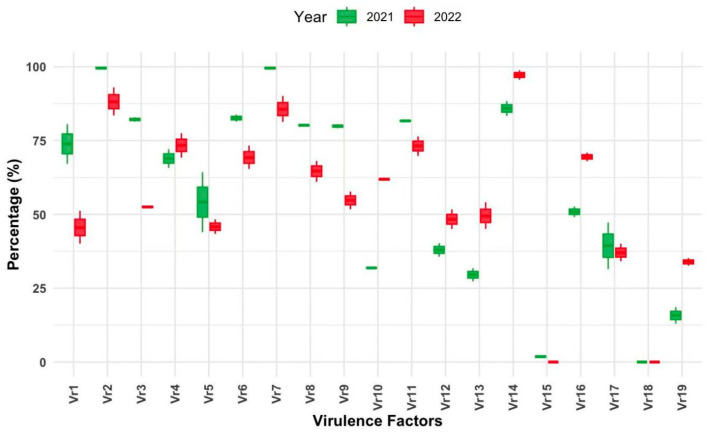
Temporal changes’ impacts on virulence factors of *Puccinia striiformis* f. sp. *tritici* isolates from Xinjiang, China during the time period 2021–2022. The virulences against Chinese differentials are: *Vr1* = Trigo-Eureka, *Vr2* = Fulhard, *Vr3* = Lutescens 128, *Vr4* = Mentana, *Vr5* = Virgilio, *Vr6* = Abbondanza, *Vr7* = Early Premium, *Vr8* = Funo, *Vr9* = Danish 1, *Vr10* = Jubilejina II, *Vr11* = Fengchan 3, *Vr12* = Lovrin 13, *Vr13* = Kangyin 655, *Vr14* = Suwon 11, *Vr15* = Zhong 4, *Vr16* = Lovrin 10, *Vr17* = Hybrid 46, *Vr18* = *Triticum spelta Album*, and *Vr19* = Guinong 22.

**Figure 2 jof-10-00870-f002:**
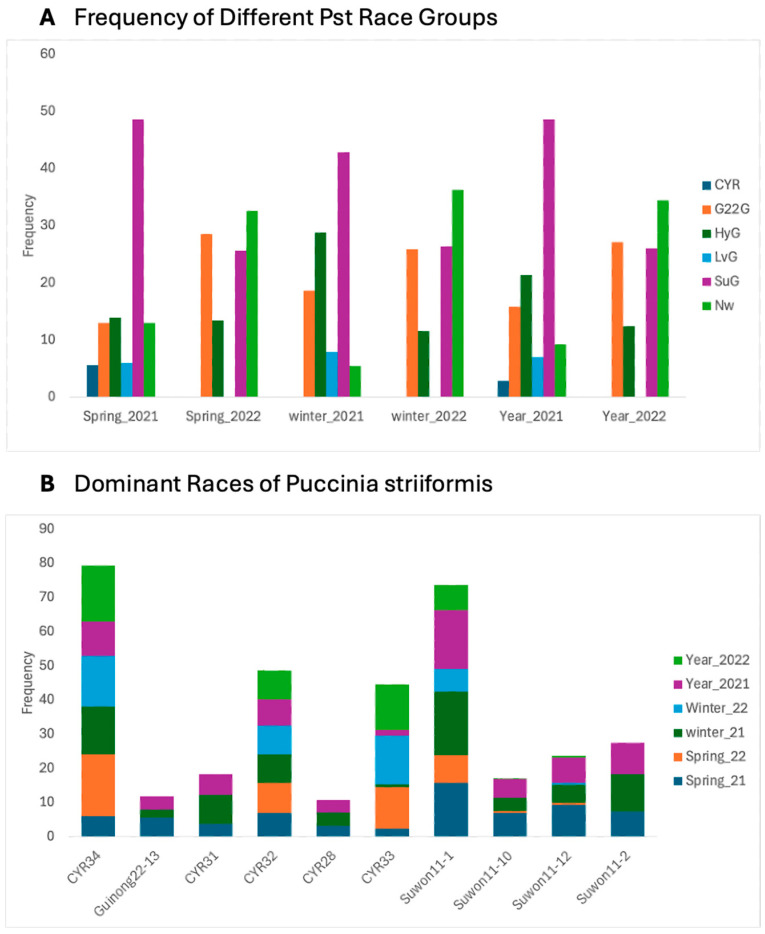
Prevalences of different races and groups of *Puccinia striiformis* in Xinjiang, China during the period 2022–2021. (**A**) Frequency of race groups, (**B**) prevalence of dominant races.

**Figure 3 jof-10-00870-f003:**
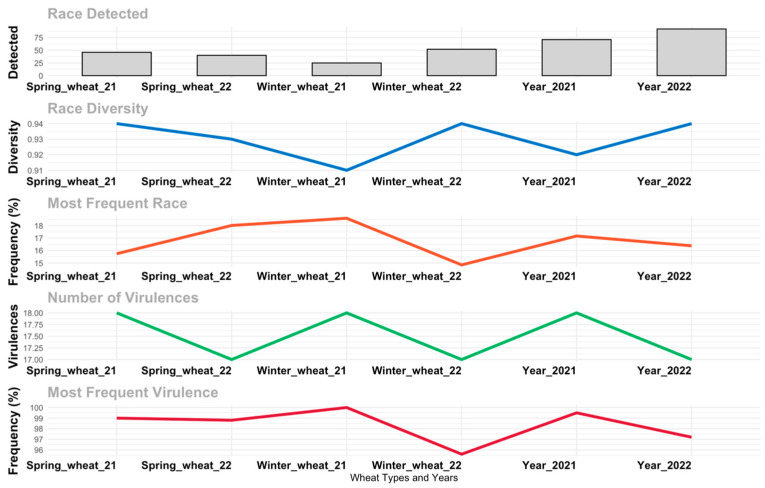
Crop season and temporal dynamic impacts on different race diversity parameters of *Puccinia striiofrmis* in Xinjiang China.

**Figure 4 jof-10-00870-f004:**
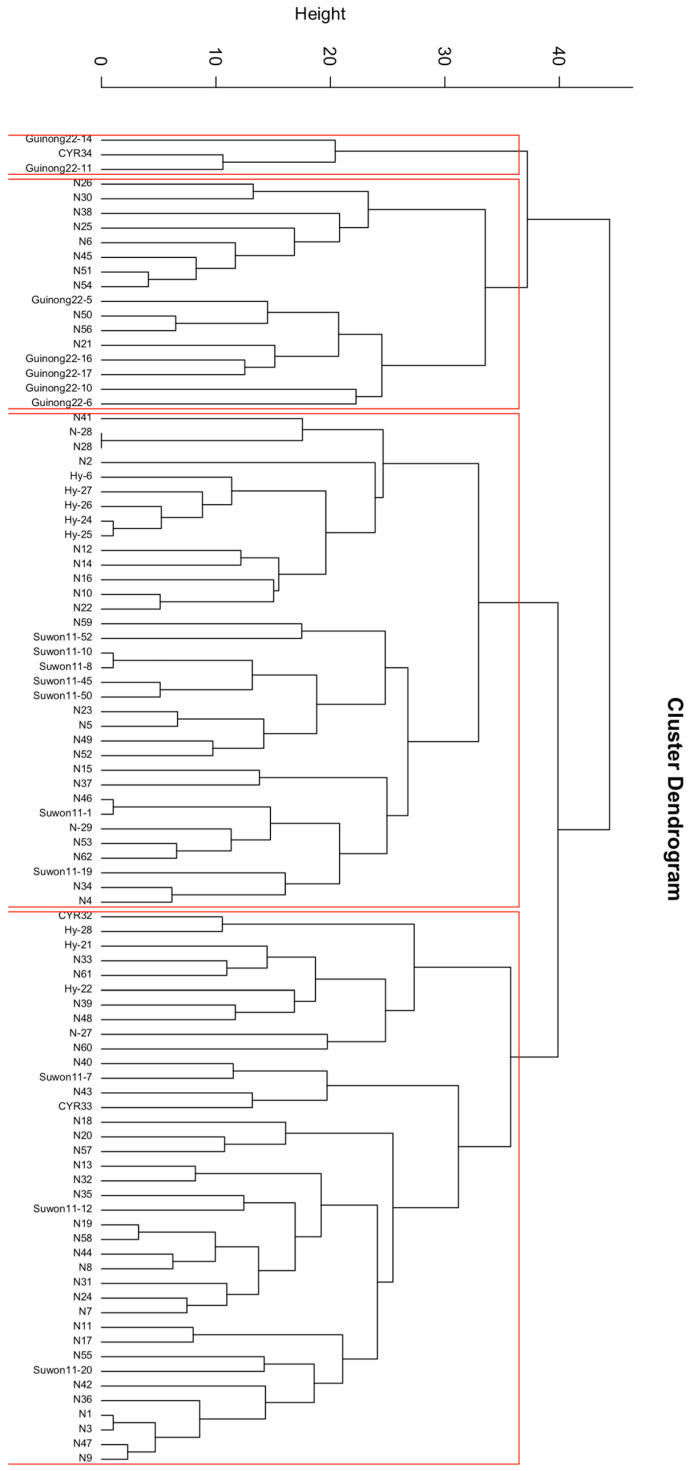
Clustering of *Puccinia striiformis* races collected from winter and spring wheat during the 2022 crop season.

**Table 1 jof-10-00870-t001:** Virulence frequencies (%) of *Puccinia striiofmris*. f. sp. *tritici* from winter wheat and spring wheat crops during the time period 2021–2022.

Virulence	Year-2022			Year-2021		
	Spring_ Wheat_22	Winter_ Wheat_22	Overall _2022	Spring_ Wheat_21	Winter_ Wheat_21	Overall_ 2021
*Vr1*	51.2	40.1	45.5	67.13	80.6	73.865
*Vr2*	93.0	83.5	88.1	99.07	100	99.535
*Vr3*	52.9	52.2	52.5	82.87	81.4	82.135
*Vr4*	69.2	77.5	73.4	65.74	72.1	68.92
*Vr5*	48.3	43.4	45.8	43.98	64.3	54.14
*Vr6*	73.3	65.4	69.2	83.8	81.4	82.6
*Vr7*	90.1	81.3	85.6	99.07	100	99.535
*Vr8*	61.0	68.1	64.7	80.56	79.8	80.18
*Vr9*	51.7	57.7	54.8	79.17	80.6	79.885
*Vr10*	61.6	62.1	61.9	31.94	31.8	31.87
*Vr11*	69.8	76.4	73.2	81.94	81.4	81.67
*Vr12*	51.7	45.1	48.3	35.65	40.3	37.975
*Vr13*	54.1	45.1	49.4	27.31	31.8	29.555
*Vr14*	98.8	95.6	97.2	83.33	88.4	85.865
*Vr15*	0.0	0.0	0.0	1.39	2.3	1.845
*Vr16*	68.0	70.9	69.5	49.07	52.7	50.885
*Vr17*	40.1	34.1	37.0	31.48	47.3	39.39
*Vr18*	0.0	0.0	0.0	0	0	0
*Vr19*	32.6	35.2	33.9	12.96	18.6	15.78

**Table 2 jof-10-00870-t002:** Frequencies of different race groups identified in both spring and winter wheat crops during the period 2021–2022.

Group *	Spring_2021	Spring_2022	winter_2021	winter_2022	Year_2021	Year_2022
*CYR*	5.57	0	0	0	2.785	0
*G22G*	12.97	28.5	18.6	25.8	15.785	27.1
*HyG*	13.88	13.37	28.8	11.54	21.34	12.43
*LvG*	6.02	0	7.9	0	6.96	0
*SuG*	48.6	25.6	42.8	26.4	48.6	26.0
*Nw*	12.98	32.6	5.5	36.3	9.24	34.5

* *CYR* (*CYR17*, *CYR23*, *CYR24*, *CYR25*, *and CYR26*), *G22G* (races virulent to differential Guinong-22), *HyG* (virulent to differential Hybrid 46), *LvG* (virulent to differential Lovrin 10), *SuG* (virulent to differential Suwon-11), and *Nw* (includes newly identified races which are low in occurrence frequency).

**Table 3 jof-10-00870-t003:** Frequencies (%) of different races of *Puccinia striifomris* f. sp. *tritici* from winter and spring wheat crops during the period 2021–2022.

Group	Races	Spring_21	Spring_22	Winter_21	Winter_22	Year_2021	Year_2022
*CYR*	*CYR17*	0.93	-	-	-	0.465	-
	*CYR23*	0.93	-	-	-	0.465	-
	*CYR24*	1.85	-	-	-	0.925	-
	*CYR25*	0.93	-	-	-	0.465	-
	*CYR26*	0.93	-	-	-	0.465	-
*G22G*	*CYR34*	6.02	18.0	14	14.8	10.01	16.4
	*Guinong22-10*	-	0.6	-	-	0	0.3
	*Guinong22-11*	-	-	-	0.5	0	0.3
	*Guinong22-13*	5.56	-	2.3	-	3.93	-
	*Guinong22-14*	1.39	7.0	2.3	4.4	1.845	5.6
	*Guinong22-16*	-	1.7	-	2.7	0	2.3
	*Guinong22-17*	-	-	-	2.2	0	1.1
	*Guinong22-5*	-	-	-	0.5	0	0.3
	*Guinong22-6*	-	1.2	-	0.5	0	0.8
*HyG*	*CYR30*	1.39	-	3.9	-	2.645	-
	*CYR31*	3.7	-	8.5	-	6.1	-
	*CYR32*	6.94	8.7	8.5	8.2	7.72	8.5
	*Hy-21*	-	-	-	0.5	-	0.3
	*Hy-22*	-	1.2	-	-	-	0.6
	*Hy-24*	-	0.6	-	0.5	-	0.6
	*Hy-25*	-	-	-	0.5	-	0.3
	*Hy-26*	-	0.6	-	1.1	-	0.8
	*Hy-27*	-	1.2	-	-	-	0.6
	*Hy-28*	-	0.6	-	-	-	0.3
	*Hy-4*	0.46	-	1.6	-	1.03	-
	*Hy-6*	0.93	0.6	4.7	0.5	2.815	0.6
	*Hy-7*	0.46	-	1.6	-	1.03	-
*LvG*	*CYR28*	3.24	-	3.9	-	3.57	-
	*CYR29*	0.93	-	0.8	-	0.865	-
	*Lovrin10-2*	0.46	-	1.6	-	1.03	-
	*Lovrin13-2*	0.93	-	1.6	-	1.265	-
	*Lovrin13-8*	0.46	-	-	-	0.23	-
*SuG*	*CYR33*	2.31	12.2	0.8	14.3	1.555	13.3
	*Suwon11-1*	15.74	8.1	18.6	6.6	17.17	7.3
	*Suwon11-10*	6.94	0.6	3.9	-	5.42	0.3
	*Suwon11-12*	9.26	0.6	5.4	0.5	7.33	0.6
	*Suwon11-13*	2.31	-	-	-	1.155	-
	*Suwon11-19*	-	-	-	1.1	-	0.6
	*Suwon11-2*	7.41	-	10.9	-	9.155	-
	*Suwon11-20*	-	-	-	1.1	-	0.6
	*Suwon11-3*	0.46	-	-	-	0.23	-
	*Suwon11-4*	-	-	0.8	-	0.4	-
	*Suwon11-45*	-	1.2	-	1.1	-	1.1
	*Suwon11-50*	-	1.2	-	0.5	-	0.8
	*Suwon11-52*	-	-	-	1.1	-	0.6
	*Suwon11-6*	0.93	-	-	-	0.465	-
	*Suwon11-7*	0.93	1.2	0.8	-	0.865	0.6
	*Suwon11-8*	2.31	0.6	1.6	-	1.955	0.3
*Nw*	N27	-	3.5	-	-	-	1.7
	N29	-	1.7	-	-	-	0.8
	N1	-	2.3	-	-	-	1.1
	N10	-	-	-	1.1	-	0.6
	N11	-	-	-	0.5	-	0.3
	N12	-	-	-	1.1	-	0.6
	N13	-	-	-	0.5	-	0.3
	N14	-	-	-	2.2	-	1.1
	N15	-	-	-	1.6	-	0.8
	N16	-	-	-	0.5	-	0.3
	N17	-	-	-	0.5	-	0.3
	N18	-	1.7	-	-	-	0.8
	N19	-	2.3	-	-	-	1.1
	N2	-	1.2	-	-	-	0.6
	N20	-	1.2	-	-	-	0.6
	N21	-	0.6	-	-	-	0.3
	N22	-	0.6	-	-	-	0.3
	N23	-	0.6	-	-	-	0.3
	N24	-	-	-	0.5	-	0.3
	N25	-	-	-	0.5	-	0.3
	N26	-	-	-	0.5	-	0.3
	N28	-	1.7	-	0.5	-	1.1
	N3	-	1.2	-	1.1	-	1.1
	N30	-	-	-	4.4	-	2.3
	N31	-	-	-	2.7	-	1.4
	N32	-	-	-	1.1	-	0.6
	N33	-	-	-	0.5	-	0.3
	N34	-	-	-	0.5	-	0.3
	N35	-	-	-	0.5	-	0.3
	N36	-	-	-	0.5	-	0.3
	N37	-	-	-	0.5	-	0.3
	N38	-	-	-	0.5	-	0.3
	N39	-	-	-	0.5	-	0.3
	N4	-	1.2	-	-	-	0.6
	N40	-	-	-	0.5	-	0.3
	N41	-	-	-	0.5	-	0.3
	N42	-	-	-	1.1	-	0.6
	N43	-	-	-	0.5	-	0.3
	N44	-	-	-	0.5	-	0.3
	N45	-	-	-	1.1	-	0.6
	N46	-	1.2	-	-	-	0.6
	N47	-	1.2	-	-	-	0.6
	N48	-	0.6	-	-	-	0.3
	N49	-	0.6	-	-	-	0.3
	N5	-	1.2	-	-	-	0.6
	N50	-	0.6	-	-	-	0.3
	N51	-	-	-	1.6	-	0.8
	N52	-	-	-	0.5	-	0.3
	N53	-	-	-	0.5	-	0.3
	N54	-	-	-	0.5	-	0.3
	N55	-	-	-	0.5	-	0.3
	N56	-	1.7	-	-	-	0.8
	N57	-	1.7	-	-	-	0.8
	N58	-	0.6	-	-	-	0.3
	N59	-	0.6	-	-	-	0.3
	N6	-	1.2	-	-	-	0.6
	N60	-	0.6	-	-	-	0.3
	N61	-	0.6	-	-	-	0.3
	N62	-	0.6	-	-	-	0.3
	N7	-	-	-	1.1	-	0.6
	N8	-	-	-	2.2	-	1.1
	N9	-	-	-	1.1	-	0.6
	Nw-1	0.93	-	-	-	0.465	-
	Nw-11	1.39	-	-	-	0.695	-
	Nw-12	0.46	-	-	-	0.23	-
	Nw-13	0.46	-	-	-	0.23	-
	Nw-14	0.93	-	-	-	0.465	-
	Nw-15	0.93	-	-	-	0.465	-
	Nw-16	0.46	-	-	-	0.23	-
	Nw-18	0.46	-	-	-	0.23	-
	Nw-19	0.46	-	-	-	0.23	-
	Nw-4	0.46	-	-	-	0.23	-
	Nw-6	0.93	-	-	-	0.465	-
	Nw-7	0.93	-	-	-	0.465	-
	Nw-8	0.93	-	-	-	0.465	-
	Nw-9	0.93	-	-	-	0.465	-
	P_10	-	-	3.1	-	1.55	-
	P_18	0.93	-	0.8	-	0.865	-
	P_20	0.93	-	0.8	-	0.865	-
	P_23	0.46	-	0.8	-	0.63	-

## Data Availability

The original contributions presented in the study are included in the article/Appendix A, further inquiries can be directed to the corresponding authors.

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
