# Peer review of "Puccinia striiformis f. sp. tritici Exhibited a Significant Change in Virulence and Race Frequency in Xinjiang, China"

_jof, 2024, doi:10.3390/jof10120870_

Round 1
Reviewer 1 Report
Overall, the study entails complex analyses and valuable data. However, I have some remarks regarding the methods employed and consequently the interpretation of results.
I strongly recommend that the authors read the following papers: Diversity of Puccinia striiformis on cereals and grasses (10.1146/annurev-phyto-072910-095230) and Field pathogenomics reveals the emergence of a diverse wheat yellow rust population (DOI 10.1186/s13059-015-0590-8) and provide a broader commentary on their results.
Author Response
We express our sincere gratitude to the editor and reviewers for their consideration of our manuscript for publication. We also appreciate their time and valuable suggestions for improving the quality of our paper. The manuscript has been revised accordingly, addressing their suggestions and answering their inquiries in sequence below. The correction made in main manuscript draft with trackmark option of MS word.
Comments 1: It is not possible to discuss a "TEMPORAL PATTERN" with data limited to just two consecutive years. Similarly, it is inappropriate to talk about a "SHIFT OF VIRULENCE" based solely on two years of data, especially when using only a differential set. The title should be revised to better reflect the essence of the study, as it is currently misleading.
Response 1: First, we would like to express our gratitude to you for dedicating valuable time to reviewing our manuscript. The title has been revised to “Puccinia striiformis exhibited a significant change in virulence and race frequency in Xinjiang during 2021-2022” We have removed the temporal pattern and the shift of virulence from the title as suggested and mentioned years at end of title to avoid any misleading information.
Comments 2: Please, provide information on molecular methods used for yellow rust race identification. Put the focus on the papers of Diane Saunders.
Response 2: Thanks for suggestion information about molecular meothods as mentiond by Diane inserted in discussion part line no 268-271.
Comments 3: After all the recent studies published by Diane Saunders and others, it is clear that relying solely on differential sets has limitations. Please pay attention to the following: 1. Many of the currently used differential lines of wheat carry additional genes for resistance, some of which may remain undetected until isolates of distant evolutionary origin are applied. The presence of extra resistance genes might hinder the genetic interpretation of results concerning virulence and avirulence genes, thereby affecting the virulence phenotype. 2. An added set of near-isogenic lines in an Avocet background has proven beneficial for distinguishing races. However, certain isolate groups, as observed in unpublished results by M.S. Hovmøller, fail to infect the susceptible control, Avocet S. Given that the samples were collected from seven countries, there is no mention here of production conditions or differences between them. Moreover, I have not seen any information about the material from which the samples were taken. From our experience, these are critical factors that must be considered to even begin discussing a shift in race composition anywhere.
Response 3: First, we would like to express our gratitude for your detailed review and suggestions. In this study, we utilized the Chinese differential lines. The Chinese race system is based on the Chinese differentials, these lines carrying multiple resistant genes. However, we also acknowledge that this system may also certain limitations. Therefore, we propose a set of near-isogenic lines of Avocet and molecular method mentioned by Diane Saunders that could be beneficial. This approach will be explored in our future studies. details we added in discussion part in line 247-271.
Comments 4: Strongly recommend that the authors read the following papers: Diversity of Puccinia striiformis on cereals and grasses (10.1146/annurev-phyto-072910-095230) and Field pathogenomics reveals the emergence of a diverse wheat yellow rust population (DOI 10.1186/s13059-015-0590-8) and provide a broader commentary on their results.
Response 4: Thanks for suggestions these two articles, which helpful to modify the overall discussion part manuscript the information related with these two emerge in discusison part incontext with our reuslt in line no. 247-271.

Reviewer 2 Report
The study evaluated the evolution of races of striped rust over time, monitoring them, which is a challenging task, as selection pressure can lead to their rapid emergence. This is of fundamental importance for management programs, as the use of resistant varieties remains the most effective method for controlling striped rust in wheat.
In general, the text requires only minor adjustments.
The study evaluated the evolution of races of striped rust over time, monitoring them, which is a challenging task, as selection pressure can lead to their rapid emergence. This is of fundamental importance for management programs, as the use of resistant varieties remains the most effective method for controlling striped rust in wheat.
It is suggested that the term 'diversity' be replaced with 'frequency' in the title.
I am uncertain whether the term 'virulence factor' is appropriately used in certain parts of the text. I suggest reviewing the concept in the highlighted sections.
The manuscript appears to be somewhat brief for an article-format paper.
In general, the text requires only minor adjustments.
In several instances, the correct spelling of the fungus (f. sp.) is written with capital letters (Tritici), which should be corrected to lowercase (tritici).
There are questions raised in the PDF text (in the form of comments) that should be addressed or clarified.
The methodology section requires further clarification and should be more thoroughly described (please refer to the observations).
The quality of Figure 4 should be improved.
The authors mention genetic changes in the text, although no such analysis was conducted. Do they plan to conduct genomic studies?
The discussion appears brief and could be expanded, particularly by addressing the issue of climate change and other factors that contribute to the emergence of new races.
Finally, the references should be more consistently standardized, and the DOI link should be included wherever possible.

Author Response
We express our sincere gratitude to the editor and reviewers for their consideration of our manuscript for publication. We also appreciate their time and valuable suggestions for improving the quality of our paper. The manuscript has been revised accordingly, addressing their suggestions and answering their inquiries in sequence below. The correction made in main manuscript draft with trackmark option of MS word.
Comments 1: The study evaluated the evolution of races of striped rust over time, monitoring them, which is a challenging task, as selection pressure can lead to their rapid emergence. This is of fundamental importance for management programs, as the use of resistant varieties remains the most effective method for controlling striped rust in wheat.
Response 1: Thanks for reviewer for highlighting the research significance and his valueable suggestions
Comments 2: It is suggested that the term 'diversity' be replaced with 'frequency' in the title. I am uncertain whether the term 'virulence factor' is appropriately used in certain parts of the text. I suggest reviewing the concept in the highlighted sections.
Response 2: Title is modified considering both reviewers remarks, the “diversity” word change to “frequency” as per suggestion.
Comments 3: The manuscript appears to be somewhat brief for an article-format paper.
In general, the text requires only minor adjustments.
In several instances, the correct spelling of the fungus (f. sp.) is written with capital letters (Tritici), which should be corrected to lowercase (tritici).
There are questions raised in the PDF text (in the form of comments) that should be addressed or clarified. The methodology section requires further clarification and should be more thoroughly described (please refer to the observations).
Response 3: The fungus name has been corrected as suggested. The raised question has been answered in the main manuscript draft and clarified it as suggested.
Comments 4: The quality of Figure 4 should be improved.
Response 4: Modified figure 4 is inserted in main draft as suggested.
Comments 5: The authors mention genetic changes in the text, although no such analysis was conducted. Do they plan to conduct genomic studies?
Response 5: The genetics changes we mentioned in refernces with previous studies that so far done in china. As our previous work, Awais et al., 2022, revealed, the genetic structure of the Puccina striiformris population in Xinjiang is distinct from other regions. However, the detailed genetic work in relation to different races has not yet been conducted in the Xinjiang region. This is a focus of our next study.
Comments 6: The discussion appears brief and could be expanded, particularly by addressing the issue of climate change and other factors that contribute to the emergence of new races.
Response 6: Discussion part expanded as suggested and issue of climatic change and other facors that contribute to emergence of new races added in disccusion . however the more details not available in context with climate and other factors. Which is our primary foucs in future study work .
Comments 7: Finally, the references should be more consistently standardized, and the DOI link should be included wherever possible.
Response 7: Doi is provided in references section.

Round 2
Reviewer 1 Report
Dear authors
Thank you for considering my comments. The manuscript is now acceptible for the publication.
There are no specific comments.